

# Changes in stigma and help-seeking in relation to postpartum depression: non-clinical parenting intervention sample

Einar B. Thorsteinsson, Natasha M. Loi and Kathryn Farr

Psychology, University of New England, Armidale, NSW, Australia

## ABSTRACT

Postpartum depression (PPD) is a prevalent mental illness affecting women, and less commonly, men in the weeks and months after giving birth. Despite the high incidence of PPD in Australia, rates for help-seeking remain low, with stigma and discrimination frequently cited as the most common deterrents to seeking help from a professional source. The present study sought to investigate PPD stigma in a sample of parents and to examine the effects of an intervention on stigma and help-seeking behaviour. A total of 212 parents aged 18–71 years ($M = 36.88$, 194 females) completed measures of personal and perceived PPD stigma and attitudes towards seeking mental health services and were randomly assigned to one of four groups: an intervention group (video documentary or factsheet related to PPD) or a control group (video documentary or factsheet not related to PPD). Results showed that there were no effects for type of intervention on either personal or perceived PPD stigma scores. No effect was found for help-seeking propensity. Males had higher personal PPD stigma than females and older age was associated with lower personal PPD stigma. Familiarity with PPD was associated with perceived PPD stigma in others but not personal PPD stigma. More work needs to be conducted to develop interventions to reduce PPD stigma in the community.

## INTRODUCTION

Mental illness has been identified as making a substantial contribution to the global burden of disease (*Prince et al., 2007*; *Vigo, Thornicroft & Atun, 2016*). According to the *World Health Organization (2017)*, depression is the leading cause of disability worldwide. Accordingly, depression and anxiety are the most prevalent mental disorders experienced by Australians (*Australian Bureau of Statistics, 2008*). The present study focuses specifically on postpartum depression (PPD). One Australian study found that at 6 months postpartum, the proportion of women reporting symptoms was 17.4% for depression, 12.7% for anxiety, and 8.1% for both anxiety and/or depression (*Yelland, Sutherland & Brown, 2010*). These proportions are even higher if the period prevalence is used (*Austin et al., 2010*; *Wynter, Rowe & Fisher, 2013*). Given that over 300,000 women give birth annually in Australia (*Australian Institute of Health and Welfare, 2018*), it is estimated that PPD affects approximately 66,000 women each year.

Corresponding author
Einar B. Thorsteinsson,
einarbt@gmail.com

Individuals with a depressive illness not only have to manage their symptoms but also cope with the stigma and discrimination these conditions receive. Stigma has a large impact on help-seeking behaviours with research indicating that many sufferers choose not to pursue treatment so as to avoid the label of 'mental illness' and the discrimination associated with such a label (*Barney et al., 2006*; *Byrne, 2000*; *Corrigan, 2004*).

## Postpartum depression

The *Diagnostic and Statistical Manual of Mental Disorders—5* (*American Psychiatric Association, 2013*) lists peripartum onset as a specifier for a major depressive episode. PPD is based on the same diagnostic criteria as a major depressive episode, but with the onset of symptoms (e.g. negative mood, sleep disturbance, significant changes in appetite, poor concentration, and loss of interest or pleasure) occurring during the last month of gestation or the first several months after delivery. The likelihood of suffering PPD is high. For example, in a study of 243 women who had recently given birth (*Jackman, Thorsteinsson & McNeil, 2017*), results found that 44% of the sample met the criteria for either *likely* (30%) or *possible* (14%) PPD as measured by the Edinburgh Postnatal Depression Scale (*Cox, Holden & Sagovsky, 1987*). PPD can range in severity from mild to severe and symptoms can begin suddenly after the birth of a child or appear gradually in the weeks or months during the first year after birth (*Bobo & Yawn, 2014*). Maternal or paternal PPD can have devastating effects on children and can lead to disturbances in children's social, behavioural, cognitive, and physical development (*Ramchandani et al., 2005, 2008*).

## Stigma

According to *Goffman (1963*, p. 3*)*, stigma refers to 'an attribute that is deeply discrediting' and, as such, is generally 'linked with illnesses or conditions that are believed to be under the individual's control or manifested as a consequence of unacceptable social behaviour' (*Pinto-Foltz & Logsdon, 2008*, pp. 21–22). When stigma occurs, a person is labelled by their illness and viewed as part of a stereotyped group (*Corrigan, 2004*). Stigmas are defined as having three elements—problems of knowledge (ignorance or misinformation, e.g. a belief that someone is dangerous); problems of attitudes (prejudice that may lead to emotional reactions, e.g. fearing someone because of the belief that someone is dangerous); and problems of behaviour (discrimination, e.g. avoiding someone with depression *Thornicroft et al., 2007*). There are also different types of stigma including personal (i.e. own beliefs about other people) and perceived (i.e. expectations of others' beliefs; *Griffiths et al., 2004*); self (i.e. stigmatising views held about the self; *Corrigan & Watson, 2002*), and structural (i.e. policies that restrict the opportunities, resources, and wellbeing of people with depression; *Corrigan, Markowitz & Watson, 2004*). Notably, personal and perceived stigma are thought to strongly influence an individual's help-seeking behaviour (*Barney et al., 2006*; *Corrigan et al., 2003*) and are the focus of the present study.

## Familiarity

Familiarity with mental illness is highly correlated with stigma. Familiarity is described as the knowledge of and experience with mental illness (*Corrigan et al., 2003*). Familiarity can

be viewed on a continuum of intensity from viewing portrayals of mental illness on television, to having a friend with a mental illness, to having a mental illness oneself. Research shows that higher familiarity reduces stigma and stereotyping towards persons with a mental illness (*Calear, Griffiths & Christensen, 2011*; *Corrigan, 2004*). Familiarity has previously been captured through recognition (a measure of knowledge of PPD) and reported as higher than 70% (*Thorsteinsson, Loi & Moulynox, 2014*).

## Stereotypes

Stigmas are also closely linked with stereotypes. Stereotypes are described as knowledge structures that the public collectively holds about a specific social group (*Corrigan, 2004*). Common stereotypes regarding people with a mental illness tend to be negative and include violence (e.g. 'people with a mental illness are dangerous'), incompetence (e.g. 'people with a mental illness are unable to look after themselves'), and blame (e.g. 'people with a mental illness are weak and are responsible for their disorder'; *Barney et al., 2006*; *Cornally & McCarthy, 2011*). Stereotyping leads to prejudice which in turn leads to discrimination towards individuals with a mental illness (*Corrigan, 2004*; *Reavley & Jorm, 2011*).

### Predictors of depression stigma

Several studies have explored common attributes of individuals who hold high levels of stigma. *Griffiths, Christensen & Jorm (2008)* found that personal stigma was higher among males and the elderly. Similarly, *Calear, Griffiths & Christensen (2011)* found that personal depression stigma was predicted by being male and having no personal or parental history (i.e. familiarity) of depression.

## Help-seeking behaviours

From a mental health perspective, help-seeking is defined as a flexible coping process that involves obtaining external assistance to treat a mental health concern (*Rickwood, Thomas & Bradford, 2013*). Despite the high incidence of mental illness in Australia, the majority of sufferers do not access professional help services. Studies show that among individuals suffering from a mental illness, up to 65% do not consult mental health professionals, resulting in a distinct variance between the prevalence of illness and the extent of professional help-seeking (*Australian Bureau of Statistics, 2008*; *Rickwood, Thomas & Bradford, 2013*).

Research shows that help-seeking behaviour may be inhibited if other members of the community are perceived as holding negative stigmas (*Corrigan & Watson, 2002*; *Dew et al., 1988*; *Vogel et al., 2007*). Recently, the topic of help-seeking behaviours has gained popularity as researchers attempt to explore and understand individuals' delayed responsive actions to symptoms of illness across a variety of health conditions.

## Challenging stigma and improving help-seeking behaviour

Postpartum depression is often left untreated as women frequently report feeling ashamed about seeking help, and hold concerns about being branded a 'bad mother' if they acknowledge that they are battling depression (*Saporito, Ryan & Teachman, 2011*).
Effective anti-stigma strategies include education (e.g. challenging the myths of mental illness with factual information); protest (e.g. making moral appeals to stop stigmatisation); and contact (e.g. creating equal interactions between the public and individuals with a mental illness; *Corrigan et al., 2003*; *Griffiths et al., 2014*). *Griffiths et al. (2004)* found that brief targeted education programs involving web-based literacy (i.e. BluePages; *The Australian National University, 2018*) and cognitive-behavioural interventions (i.e. moodgym; https://moodgym.com.au/) were effective in reducing levels of depression stigma.

To date, there has been some research interest in depression-related stigma (*Barney et al., 2006*; *Byrne, 2000*; *Griffiths et al., 2014*). However, few studies have addressed PPD stigma. Research has found that knowing someone with depression is associated with less stigmatising attitudes (*Corrigan & Watson, 2007*; *Griffiths, Christensen & Jorm, 2008*). Targeted education programs that teach individuals that mental disorders are an illness like any other have been shown to be effective (*Corrigan & Watson, 2007*).

## The present study

Given the lack of research on depression stigma, the aim of the current study was to examine existing levels of personal and perceived depression stigma among a sample of parents to determine if targeted intervention materials (i.e. factsheet and documentary) can reduce levels of depression stigma. Based on the literature discussed, it was hypothesised that (1) an education intervention would influence depression stigma and help-seeking behaviour scores. Specifically, the participants in the PPD factsheet or video documentary groups would have lower depression stigma and higher help-seeking behaviour scores (pre-test vs. post-test measures) compared to the control groups (i.e. family documentary or raising children factsheet); (2) levels of familiarity with PPD would be inversely correlated with levels of both personal and perceived stigma; and (3) males and older persons would have higher levels of personal PPD stigma than females and younger persons, respectively.

## METHODS

### Participants

The study was accessed by 594 participants. However, due to an incompatibility between the software platform and video usage on smartphones, tablets, and some web browsers, a majority of the participants ($n = 373$; 62.8%) were unable to complete the survey. A further nine participants were eliminated from the study due to incomplete questionnaires. Therefore, useable data (*Thorsteinsson, Loi & Farr, 2018*) was obtained from 212 participants (35.7% retention rate) aged between 18 and 71 years ($M = 36.88$, $SD = 8.71$). An a priori power analysis showed that 192 participants would be needed assuming a low-medium effect size (Hedges' $g$) of 0.28 (based on the findings of *Griffiths et al., 2014*), a low-medium effect size ($d$) of 0.28 was anticipated for the current study to achieve power = 0.80 (*Faul et al., 2007*).

To be included in the study, participants were required to have one or more children. The final sample comprised 18 males (8.5%) and 194 females (91.5%). A large proportion

of participants (72.1%) had one to two children, with 27.9% of participants having three or more children. The majority of participants held a bachelor's degree or higher ($n$ = 120, 56.6%). A considerable proportion ($n$ = 34; 16%) of participants had medical or allied health training. Table 1 shows demographics within each intervention group.

## Measures

The online self-report questionnaire battery consisted of 116 questions. All participants in the study completed a pre-intervention questionnaire that included a variety of sociodemographic and illness exposure measures. The sociodemographic variables collected included sex and age. The illness exposure variables measured included history of personal PPD, history of personal mental illness, family member/friend history of PPD, family member/friend history of mental illness, and current confidante/s to discuss mental health concerns. Familiarity with PPD was measured with one question 'Have you personally experienced postnatal depression?' answered either as 0 (*No*) or 1 (*Yes*).

### Personal and perceived PPD stigma

Personal and perceived PPD stigma was measured using the Depression Stigma Scale (DSS; *Griffiths et al., 2004*) with the wording modified to include the word 'postnatal' Postnatal was used rather than postpartum as the former was considered a better match to then community knowledge. The 18-item scale assesses stigmatising attitudes and beliefs towards individuals with PPD and consists of two subscales: personal stigma and perceived stigma. The *personal stigma* subscale assesses the participant's own attitudes towards individuals with PPD (nine items: e.g. 'People with postnatal depression are unpredictable'), while the *perceived stigma* subscale assesses the participant's perception of the attitudes of others to individuals with PPD (nine items: e.g. 'Most people believe that people with postnatal depression are unpredictable'). Items on each of the subscales are rated on a five-point Likert scale ranging from 0 (*strongly disagree*) to 4 (*strongly agree*). A total scale score is calculated by summing item scores, with total scale scores ranging from 0 to 72. Higher scores are indicative of greater stigma. The two subscales have been reported as having moderate to high internal consistency ($\alpha$ = 0.72–0.82; *Griffiths, Christensen & Jorm, 2008*; *Griffiths et al., 2004*), and moderate test–retest reliability (*Griffiths et al., 2004*). In the present study, a moderate level of internal consistency was attained, with a McDonald's omega of 0.81 for personal PPD stigma and 0.90 for perceived PPD stigma.

## Attitudes towards help-seeking

Attitudes towards help-seeking for treatment of a mental health issue was assessed using the Inventory of Attitudes towards Seeking Mental Health Services Scale (IASMHS; *Mackenzie et al., 2004*). The IASMHS is a 24-item instrument that measures beliefs about seeking professional help for mental health problems. Items are presented on a five-point Likert scale ranging from 0 (*strongly disagree*) to 4 (*strongly agree*), with high scores reflecting more positive attitudes. The scale comprises three subscales: psychological openness (eight items: e.g. 'Psychological problems, like many things, tend to work out by themselves'); help-seeking propensity (eight items: e.g. 'I would

**Table 1 Participant demographics and manipulation checks.**

| Measure | Condition | | | | Statistical comparison |
|---|---|---|---|---|---|
| | Experimental video documentary | Control video documentary | Experimental factsheet | Control factsheet | |
| $n$ | 42 | 50 | 58 | 62 | $\chi^2(3, N = 212) = 4.45, p = 0.217$ |
| Sex ($n$ males/females) | 5/37 | 5/45 | 2/56 | 6/56 | $\chi^2(3, N = 212) = 2.79, p = 0.426$ |
| Age ($M$ (SD)) | $39.88_a$ (11.02) | $36.70_{ab}$ (9.08) | $35.02_b$ (7.42) | $36.73_{ab}$ (7.32) | $F(3, 212) = 2.62, p = 0.052$ |
| Education ($n$) | 4/6/6/19/7 | 2/4/13/22/9 | 3/8/18/18/11 | 3/5/20/22/12 | $\chi^2(4, N = 212) = 5.94, p = 0.203$ |
| Medical or allied health training ($n$ Yes/No) | 6/36 | 8/42 | 8/50 | 12/50 | $\chi^2(3, N = 212) = 0.82, p = 0.845$ |
| Intervention improved: ($M$ (SD)) | | | | | |
| Knowledge of PPD | $4.18_a$ (0.68) | $2.26_b$ (1.26) | $4.04_a$ (0.93) | $3.23_c$ (1.18) | $F(3, 196) = 33,13, p < 0.001$ |
| Likelihood seek professional help | $4.00_a$ (0.93) | $2.37_c$ (1.32) | $3.71_a$ (1.17) | $3.06_b$ (1.20) | $F(3, 180) = 23.12, p < 0.001$ |
| Likelihood encourage others to seek professional help | $4.32_a$ (0.78) | $2.47_c$ (1.42) | $4.29_a$ (0.65) | $3.45_b$ (1.19) | $F(3, 176) = 29.37, p < 0.001$ |

**Note:**

Parameter estimates in each row that share a subscript do not differ significantly (Sidak post hoc test). Education coded as 1 = year 10 or below, 2 = year 12 or HSC, 3 = TAFE, 4 = University or college degree, and 5 = postgraduate.

want to get professional help if I were worried or upset for a long period of time'); and indifference to stigma (eight items: e.g. 'Having been mentally ill carries with it a burden of shame'). Higher total scores indicate a more positive attitude towards seeking help. In the current study, a moderate level of internal consistency was attained based on McDonald's omega, with 0.77 for psychological openness, 0.76 for help-seeking propensity, and 0.83 for indifference to stigma.

## Intervention Groups

Using Qualtrics software, participants were randomly assigned to one of four intervention groups, see Fig. 1.

### Experimental factsheet group

Participants in this group were shown a digital factsheet written by the Post and Antenatal Depression Association (PANDA) and used with permission from the Chief Executive Officer (B. Horton, 2014, personal communication). The factsheet spanned two pages and contained information outlining the contributing factors, common symptoms, and potential treatments of PPD.

### Control factsheet group

Participants in this group were shown an electronic factsheet compiled by the Raising Children Network (2006) and used with permission from the network (Raising Children Network, 2014, personal communication). The factsheet contained information about parenting, the challenges of being a parent, and tips to deal with parenting difficulties. The factsheet was designed to be generic, therefore to control for the effects of the term PPD used in the experimental factsheet group, the control group's factsheet did not contain any references to PPD.
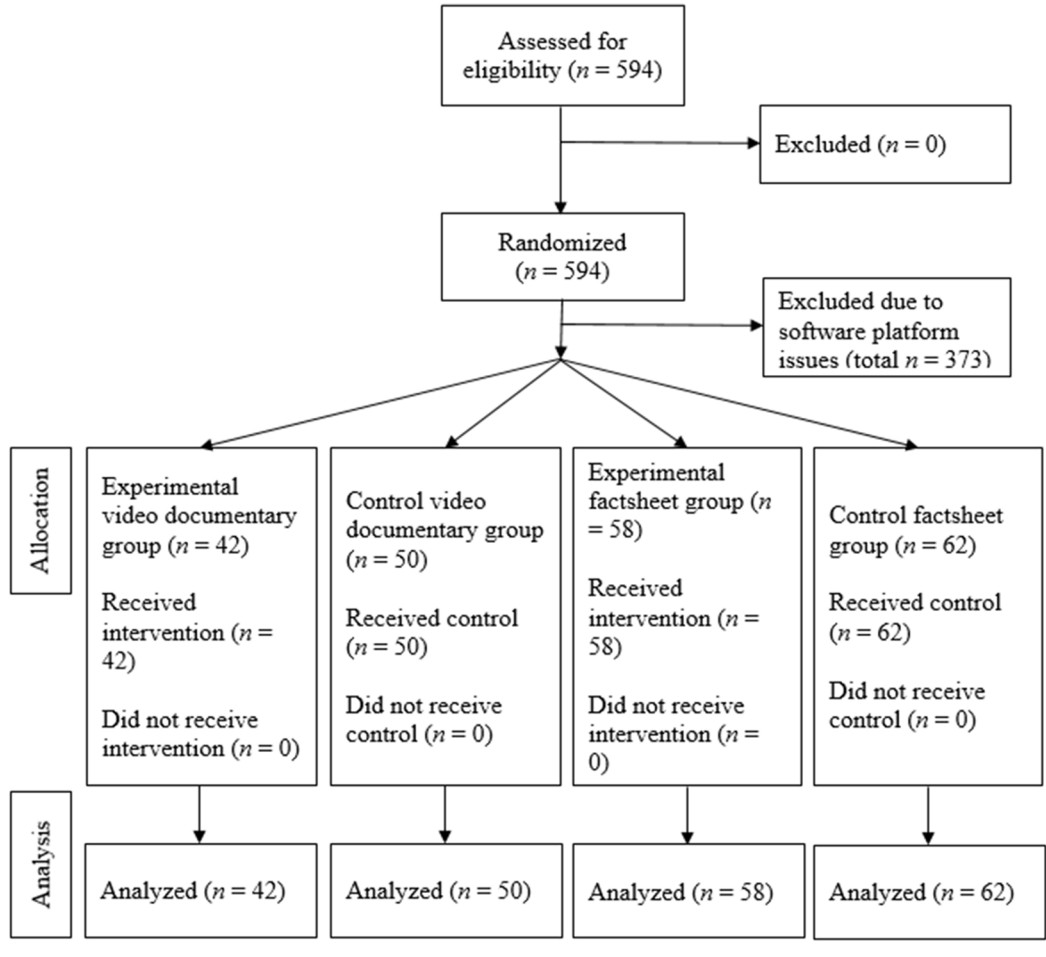

**Figure 1 Flow diagram showing randomization of participants.**

### Experimental documentary group

Participants in this group were required to view a documentary produced by PANDA and used with permission (B. Horton, 2014, personal communication). The documentary ran for 8 min 38 s and outlined the burden of PPD, gave real-life experiences of sufferers of PPD and their families, and included insights into the illness from experts and health professionals.

### Control documentary group

Participants in this group were required to view a documentary comprising five 'Meet the Families' video clips taken from the Raising Children Network website (www.raisingchildren.net.au). These clips were used with permission (Raising Children Network, 2014, personal communication). The duration of the documentary was 7 min 55 s and depicted the stories of five families and their experiences of raising children. To control for the effects of the term PPD used in the experimental documentary group, this documentary did not contain any references to PPD.

## Procedure

Ethical approval for the study was obtained from the University of New England's Human Research Ethics Committee (Approval Number: HE14-154). A pilot study was implemented to check for clarity of questions and the effects of question order. Three questions were reworded when they were identified as being difficult to comprehend. No order effects were found for question order and thus question presentation remained unchanged.

Potential participants were recruited on social networking sites (e.g. Facebook), online forums (e.g. Reddit), and the university's online learning platform. The study was administered through Qualtrics (Provo, UT, USA), a secure online survey site. Participants were provided with an information sheet fully informing them of the purpose of the study. They were then advised that their responses were anonymous and that withdrawal from the study was permitted at any time without consequence. Participants were also informed that activation of the 'Proceed to study' button constituted their informed consent.

Participants were asked demographic and illness exposure questions before the DSS and IASMHS were presented. Employing a randomised control trial study design, participants were randomly assigned to one of the four intervention groups. At the conclusion of the intervention, participants were once again required to complete the DSS and IASMHS. At the close of the survey, participants were thanked for their time and invited to enter an optional prize draw for a chance to win an AUD$50 iTunes gift card. Participants were then redirected to the PANDA website where they could find further information about PPD.

## Manipulation checks

At the end of the intervention, all participants were asked three questions about the intervention: 'How effective were the resources in improving your knowledge and understanding of postnatal depression?', 'How effective were the resources in improving the likelihood that you would seek professional help for a psychological problem?', and 'How effective were the resources in improving the likelihood that you would encourage others to seek professional help for a psychological problem?'. These questions were answered on a scale from 1 (*resources were poor*) to 5 (*resources were excellent*). The two experimental groups were rated significantly higher than the two control groups on all three questions by the participants, see Table 1.

## Statistical analysis

Statistical analyses were performed using SPSS (versions 24 and 25) and JASP (version 0.8.6.0). Missing values were replaced using the replace missing values, series mean method. Selected questions were reverse scored in line with measurement requirements. As all assumptions of normality were met, raw data was employed in the following analyses. A one-way analysis of covariance (ANCOVA) was used to examine the impact of intervention groups on personal and perceived PPD stigma and help-seeking behaviour scores. Pre-intervention levels of PPD stigma and help-seeking were employed as covariates.

**Table 2 Means and standard deviations for assessment of postpartum stigma by experimental condition.**

| Measure | Condition | | | |
|---|---|---|---|---|
| | Experimental video documentary | Control video documentary | Experimental factsheet | Control factsheet |
| $n$ | 42 | 50 | 58 | 62 |
| Pre-intervention personal | 6.21 (4.37) | 7.12 (4.81) | 6.48 (4.38) | 6.92 (4.01) |
| Post-intervention personal | 6.31 (4.36) | 7.39 (4.81) | 7.79 (4.64) | 7.97 (4.47) |
| Pre-intervention perceived | 17.64 (5.05) | 18.04 (7.01) | 16.48 (7.58) | 16.81 (6.59) |
| Post-intervention perceived | 18.29 (5.33) | 18.90 (6.57) | 16.08 (7.77) | 17.65 (6.71) |
| Pre-intervention help-seeking propensity | 2.96 (0.57) | 2.87 (0.52) | 2.96 (0.54) | 2.87 (0.49) |
| Post-intervention help-seeking propensity | 3.04 (0.48) | 2.85 (0.55) | 2.97 (0.65) | 2.89 (0.53) |

## RESULTS

### Intervention

The ANCOVA showed that following the intervention, personal PPD stigma scores differed between the four groups, $F(3, 207) = 3.05$, $p = 0.030$, partial $\eta^2 = 0.04$. However, there were no statistically significant pairwise comparison effects. Furthermore, Table 2 shows that the intervention did not reduce personal PPD stigma.

The pattern of findings for perceived PPD stigma did not indicate a statistically significant difference, $F(3, 207) = 2.19$, $p = 0.090$, partial $\eta^2 = 0.03$. There were no significant pairwise comparison and perceived PPD stigma increased from pre- to post-test, see Table 2.

The results for help-seeking propensity were not affected by the intervention, $F(3, 207) = 1.15$, $p = 0.332$, partial $\eta^2 = 0.02$, see Table 2.

### Sex, age, and familiarity with PPD

Post-intervention personal PPD stigma was used in the following analysis, but results were almost the same for pre-intervention personal PPD stigma. Males had higher personal PPD stigma ($M = 10.11$, SD = 4.75) than females ($M = 7.21$, SD = 4.50), $t(210) = 2.61$, $p = 0.010$, Hedges' $g = 0.64$, 95% CI [0.15–1.13]. Age was associated with personal PPD stigma, $r(210) = -0.18$, $p = 0.010$, thus the older the participant the less their personal PPD stigma. Familiarity (no = 0, yes = 1) with PPD was correlated with post-intervention perceived PPD stigma scores, $r(210) = 0.21$, $p < 0.01$ but not with post-intervention personal PPD stigma scores, $r(210) = 0.01$, $p = 0.839$. Reclassifying familiarity to include both personal familiarity and familiarity through friends or relations also resulted in low correlations ($r = -0.01$ and $r = 0.06$, respectively). A multiple regression with sex, age, and familiarity as predictors explained only 3.2% ($R^2$) of the variance in post-intervention personal PPD stigma scores, $F(3, 208) = 2.31$, $p = 0.077$. The same predictors only explained 0.6% ($R^2$) in post-intervention perceived PPD stigma scores.

## Post hoc analysis

An additional analysis examining the relationship between personal familiarity and help-seeking was conducted. Results revealed a significant effect of familiarity on attitudes towards help-seeking scores, $F(1, 209) = 4.31$, $p = 0.039$, partial $\eta^2 = 0.02$. Participants with a personal experience of PPD had more negative attitudes towards help-seeking ($M = 64.39$, SD = 12.28) compared to participants with no personal experience of PPD ($M = 68.97$, SD = 12.37).

## DISCUSSION

This study sought to examine how an education intervention would impact PPD stigma and, more indirectly, help-seeking propensity.

### Education intervention and help-seeking

It was hypothesised that an education intervention would significantly influence PPD stigma scores. Specifically, that participants in the intervention groups (factsheet or video) would have lower personal and perceived PPD stigma and higher help-seeking scores compared to the control groups. Results, however, revealed that the intervention had no significant effect on PPD stigma scores. These findings are inconsistent with previous findings (*Corrigan et al., 2003*; *Griffiths et al., 2004*) demonstrating that educational interventions could significantly reduce personal and perceived stigma. One reason for these inconsistent findings is that previously used interventions tended to be more interactional. That is, participants were required to perform a type of action. As participants in the present study were simply required to watch a video or read a factsheet, they may not have engaged as much with the material. Another suggestion for these inconsistent findings is that group sizes in previous studies tended to be much larger ($n = 165$; *Griffiths et al., 2004*) than the present study ($n = 53$). Future studies may benefit from including several overt manipulation checks throughout the survey to confirm that the intervention groups were successfully manipulated.

We anticipated that viewing a PPD documentary would lead to increased help-seeking behaviours. Results showed that there was no significant effect of type of intervention on help-seeking propensity.

### Sex and age

Consistent with previous research (*Calear, Griffiths & Christensen, 2011*; *Corrigan, 2004*), males had higher levels of personal PPD stigma than females. Additionally, we hypothesised that older persons would have higher levels of personal stigma, however, this was not supported. While age was associated with personal PPD stigma, the results indicated that the older the participant, the *lower* their levels of personal PPD stigma. This result is inconsistent with previous research conducted by *Corrigan et al. (2003)* who found that older persons exhibited more personal stigma towards individuals with depression. One explanation for these varied findings is that the older aged participants in the current study may have had high levels of familiarity with PPD. It is quite possible that their own child may have experienced PPD and this may have influenced their personal PPD stigma levels.

## Familiarity

Inconsistent with previous research, familiarity was positively correlated with perceived PPD stigma scores rather than inversely (*Corrigan, 2004*; *Griffiths et al., 2004*). Unexpectedly, participants who had personally experienced PPD had significantly higher perceived stigma scores. The results of the current study suggest that individuals who have experienced PPD expect that other people will hold higher levels of stigma towards people with PPD. Perceived stigma has been shown to negatively affect help-seeking behaviours (*Corrigan & Watson, 2002*). If those experiencing PPD do anticipate high levels of stigma in the community, this could greatly impact their help-seeking behaviours, leading to cases of PPD being left undiagnosed and untreated. An alternative explanation for these results could be that individuals who have experienced PPD are quite sensitive about the topic and unwittingly overinflated the expected negative attitudes of others.

Familiarity with PPD was not associated with the participants' own stigma towards people with PPD (personal PPD). It is unclear why familiarly with PPD does not reduce the participants' own stigma towards PPD. Any relationship between familiarly with PPD and personal PPD stigma may be 'hidden' by the type of familiarity, thus future research may want to ask questions that assess the type of familiarity in more detail than in the present study.

## Limitations and future research

The participants were predominantly female and educated and this should be taken into account when considering generalising the findings to the broader community. A larger sample of males is needed to enable a reliable comparison with females given that males tend to have a worse mental health literacy than females (*Gibbons, Thorsteinsson & Loi, 2015*). The sample is also biased towards participants with medical or allied health training (i.e. 16%), potentially as a result of a low retention rate. Furthermore, future studies might like to sample from a population of pregnant women or women wanting to be pregnant and their partners as they are the population at risk of PPD.

Further research is needed to fully understand the different types of stigma that are associated with PPD and any social, sex, and age factors that may underlie it. Research of a longitudinal nature is also required to investigate if changes in attitudes lead to changes in behaviour. Of specific interest is whether a targeted intervention can effectively reduce personal and/or perceived stigma levels and lead to a change in behaviour towards individuals with PPD. Such interventions could include clear hospital guidelines such as have been successful in increasing individuals' psychological wellbeing (*Basile & Thorsteinsson, 2015*).

## CONCLUSIONS

It is apparent that stigmatising beliefs and attitudes regarding PPD do exist in the parenting community. Stigma towards people with PPD (personal stigma) was predicted by being male and being younger. Personal familiarity with PPD increased people's expectations that there would be stigma towards those with PPD. The stigma intervention was not effective in the present study suggesting that such interventions need to be

 

developed differently. It may be appropriate to develop a broader-based education program targeting attitudes, disorder characteristics, and effects of stigma for the general parenting community.

### Funding
The authors received no funding for this work.

### Competing Interests
The authors declare that they have no competing interests.

### Author Contributions
- Einar B. Thorsteinsson conceived and designed the experiments, analysed the data, prepared figures and/or tables, authored or reviewed drafts of the paper, approved the final draft.
- Natasha M. Loi authored or reviewed drafts of the paper, approved the final draft.
- Kathryn Farr conceived and designed the experiments, performed the experiments, analysed the data, authored or reviewed drafts of the paper, approved the final draft.

### Human Ethics
The following information was supplied relating to ethical approvals (i.e. approving body and any reference numbers):

The University of New England, Australia granted ethical approval to carry out the study (Approval number: HE14-154).

### Data Availability
Thorsteinsson, Einar; Loi, Natasha; Farr, Kathryn (2018): Intervention for reducing stigma and increasing help-seeking in postpartum depression: Non-clinical parenting sample. figshare. Dataset. https://doi.org/10.6084/m9.figshare.6234848.v3.

### Supplemental Information
Supplemental information for this article can be found online at http://dx.doi.org/10.7717/peerj.5893#supplemental-information.

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
