# Peer review of "Changes in stigma and help-seeking in relation to postpartum depression: non-clinical parenting intervention sample"

_PeerJ, doi:10.7717/peerj.5893_

## Round 0.1 · original submission · Major Revisions

The two reviewers made a number of important points for you to address. Please respond to each point raised to explain how you have addressed their concern or why you disagree. Overall, better choice of references and more detailed reporting of methods would greatly strengthen the paper. I agree with Reviewer 2 that the paper should be reported according to the CONSORT statement, as it is a randomised controlled trial of an intervention (see http://www.equator-network.org/reporting-guidelines/consort/).

Reviewer 1 ·

Basic reporting

The basic reporting is clear and the English is quite good. There are major problems with the references, and some problems with the article structure, including the fact that I believe that two more Tables should be presented in addition to the single Table provided. More specifics are provided in General Comments below. The results are relevant to the hypotheses.

Experimental design

The aims of the study are appropriate to the Journal.
Research question is well-defined and meaningful, and is important to the topic at hand-mental health stigma and receipt of care in general, and in the case of postpartum depression in particular. There are some problem with the methods, as noted below.

Validity of the findings

The findings are clear even if negative. The conclusions are appropriate, although selection bias and its implications should be discussed in more detail.

Additional comments

This is an important issue and interesting approach to studying the problem of mental health stigma and help-seeking behaviors. Some corrections and changes could improve the manuscript so that this could be a meaningful contribution to the professional literature.

Here are more specific comments:

Abstract:
Line 33-34: the word 'on' is missing before 'stigma'
Introduction:
• Several of the references provided are not appropriate for peer review journals. Organization websites that make statements about rates and associated variables are not acceptable, as they are not results of published peer-reviewed research. I refer to the beyondblue, PANDA, Gidget Foundation. The statements made in the text should be backed up by serious research.
• Line 68: the statement regarding a PPD rate of 44% is far different than that reported in most large-scale studies in developed countries, and is not a fair reflection of research findings. Further the research referenced (Jackman et al.) has a clear selection bias, since recruitment was done via social media PPD support groups, etc., which are much more likely to attract women who are suffering depressive symptoms. Thus the rate reported is not generalizable. Further, the use of this rate and reference suggests a lack of the researchers' knowledge regarding the body of epidemiological studies of PPD.
• Line 76-79: two references seem to be provided for a single quote. Please clarify the correct reference.
• Line 84-88: As noted above, beyondblue is not an appropriate source for the statements. Mentions of 'stigma' on that website are mainly in describing their efforts or in the Forums..
• Presentation of the references is not consistent--sometimes et al. is used, sometimes all authors are noted; the editorial rules should be followed. Also the use of parentheses when noting authors is confusing--sometimes the author's name is within the parentheses even though the names are part of the text (eg., line 109, 134 etc.).
• Line 120: the reference to Australian Bureau of Statistics is too general--I could not find the data at the url indicated in the References. The information should be more specific.
• Line 136--the reference Australian National University is not on the Reference list; to what research is this referring?
Method:
• Participants: Better description of the participants should be presented, as well as the differences in characteristics between the various study groups (with significance levels)--best presented as a Table.
• Line 169--16% of the participants having medical training is not a 'small proportion' as this is likely much greater that the proportion in the general population. Thus raising questions about the recruitment and potential bias of the study population, which should be addressed later in the limitations. This is also true of the very low retention rates (35.7%).
• Line 175 'previous allied health training'--why is this included and not discussed later on.?
• Line 224--the word 'to' is missing before PPD.
Procedure:
• Line 238--Please name the University.
• Line 258--was the question posed to all groups (intervention and control)?
• Line 264-266--this may be better placed in the Results section.
Results:
• Line 268--the Statistical Analysis section is usually placed in the Methods section.
• Lines 286-295--A Table with these results would be easier to follow and understand.
Table 1: The scale name "Postpartum Depression Sale" is not the same as that mentioned in the text "Depression Stigma Scale."

·

Basic reporting

My main recommendation is that this study should be reported according to the CONSORT Statement.
Has the trial been registered? If so, give details. If not, it needs retrospective registration.
The Abstract needs to specify what the interventions involve.
Lines 97-98. This sentence is about recognition, which is not the same as familiarity.
Line 119. I think this should say “prevalence” rather than “incidence”.
Lines 163-164. In reporting the power analysis, please give the effect size that was the basis of the calculation and how this effect size was chosen.
The authors use Cronbach’s alpha as an index of reliability. The consensus of opinion among psychometricians these days is that alpha has major limitations and should be replaced by McDonald’s omega. See https://www.ncbi.nlm.nih.gov/pubmed/28557467 and https://www.ncbi.nlm.nih.gov/pubmed/24844115 .
In examining the correlates of stigma (lines 287-295), univariate methods are used. Given that the correlates may not be independent, this should also be done with multiple regression.
I could not understand the sentence on lines 298-300.

Experimental design

No comments.

Validity of the findings

The choice of sample for this study is not ideal, as the participants are not necessarily at risk of PPD (e.g. some are elderly). The study would be better if carried out with pregnant women and their partners, or at least people planning to be parents in the future. This limitation needs acknowledging.

Additional comments

None.

---

## Round 0.2 · accepted · Accept

Both reviewers agreed that you had addressed their concerns adequately.

# Reviewer 1 ·

Basic reporting

This is a re-review; issues previously noted have been addressed.

Experimental design

Issues noted in the original review have been adequatly addressed

Validity of the findings

The findings are clear. Bias has been adequately noted.

Additional comments

The issues in the review have been adequatey addressed.

·

Basic reporting

No comment.

Experimental design

No comment.

Validity of the findings

No comment.

Additional comments

No comment.